# Experimental Determination Influence of Flow Disturbances behind the Knife Gate Valve on the Indications of the Ultrasonic Flow Meter with Clamp-On Sensors on Pipelines

**DOI:** 10.3390/s23104677

**Published:** 2023-05-11

**Authors:** Piotr Piechota, Piotr Synowiec, Artur Andruszkiewicz, Wiesław Wędrychowicz, Elżbieta Wróblewska, Andrzej Mrowiec

**Affiliations:** 1Department of Thermal Science, Faculty of Mechanical and Power Engineering, Wrocław University of Science and Technology, Wybrzeze Wyspianskiego 27, 50-370 Wrocław, Poland; piotr.synowiec@pwr.edu.pl (P.S.); artur.andruszkiewicz@pwr.edu.pl (A.A.); wieslaw.wedrychowicz@pwr.edu.pl (W.W.); e.wroblewska@pwr.edu.pl (E.W.); 2Polytechnic Faculty, Calisia University, 62-800 Kalisz, Poland; a.mrowiec@akademiakaliska.edu.pl

**Keywords:** ultrasonic flow meter, clamp-on, flow disturbance, correction factor measurement error, knife gate valve, laser doppler anemometry

## Abstract

The aim of this work is to experimentally determine and evaluate the value of the correction factor for ultrasonic flow meters in order to improve their accuracy. This article concerns flow velocity measurement with the use of an ultrasonic flow meter in the area of disturbed flow behind the distorting element. Clamp-on ultrasonic flow meters are popular among measurement technologies due to their high accuracy and easy, non-invasive installation, because the sensors are mounted directly on the outer surface of the pipe. In industrial applications, installation space is usually limited and, therefore, flow meters frequently have to be mounted directly behind flow disturbances. In such cases, it is necessary to determine the value of the correction factor. The disturbing element was a knife gate valve, a valve often used in flow installations. Water flow velocity tests were performed using an ultrasonic flow meter with clamp-on sensors on the pipeline. The research was performed in 2 series of measurements with different Reynolds numbers of 35,000 and 70,000, which correspond to a velocity of approximately 0.9 m/s and 1.8 m/s. The tests were carried out at different distances from the source of interference, within the range of 3–15 DN (pipe nominal diameter). The position of the sensors at successive measurement points on the circuit of the pipeline was changed by 30 degrees. Flow velocity measurements were carried out for two different levels of the valve’s closure: 1/3 and 1/2 of the valve’s height. For the collected velocity values at single measurement points, the values of the correction coefficient, K, were determined. The results of the tests and calculations prove that compensation error of measurement performed behind the disturbance without keeping the required straight sections of the pipeline is possible by using the factor K*. The analysis of the results made it possible to identify the optimal measuring point at a distance from the knife gate valve as being smaller than specified in the standards and recommendations.

## 1. Introduction

Measurement of the flow rate is one of the most important measurements in the engineering industry. There is a special necessity for mass and volume flow measurements in the power, chemical and petrochemical industry as well as in many other engineering areas such as building or environmental engineering. The parameter measured in the power industry might be the flow rate of cooling water in the Clausius–Rankine cycle. In the field of environmental engineering, information concerning the medium in air condition systems, ventilation or heating systems is highly significant. Defining the medium flow rate is fundamental in order to perform a mass balance as well as the system’s energy balance.

The examples given above show the variety and scope in engineering fields which require flow rate measurements. Continuous technological development entails the need to perform measurements with high accuracy. To ensure the measurements of velocity and flow stream are the most precise as possible is especially important, because this information is usually widely used in the calculation process as well as during the functioning of a control system of a given installation. The uncertainty of the flow rate measurement, in accordance with the uncertainty propagation, affects the final uncertainty, which in turn burdens the evaluation of the thermo-flow balance as well as the determination of the value of the mass, entropy, enthalpy and other physical and thermodynamic parameters. The results of the measurements can be used for the appropriate management of the technological process. They can also be the basis for making settlements for the consumption of a given medium, i.e., the calculation of fees for water consumption.

The display accuracy of a specific measurement instrument which is determined by its characteristic parameters, such as the limit error and resolution, is one of many factors contributing to the uncertainty budget for measurements performed with the usage of this instrument in a given installation as well as in given conditions of measurement. The uncertainty budget for a measurement also consists of components connected to the distribution of the measured quantity (uncertainty type A u_A_) as well as components connected to the accuracy of the measuring instrument installation and the conditions of performing a measurement (components of the uncertainty type B u_B_) [1,2,3]. Apart from the above-mentioned classic method for the determination of the uncertainty budget based on the uncertainty propagation law, there is a possibility to determine the uncertainty budget with the use of the numerical Monte Carlo method [4,5]. The possibility of mounting during the uninterrupted work continuity in the system, influence on the processes occurring in the system (non-invasiveness) and economic factors are the most important elements which determine the use of a measuring device. There are several highly desirable features of flow meters:High measurement accuracy—high sensitivity and resolution;Non-invasiveness during operation;Non-contact;Service life—long-term operation;Independence from environmental conditions/system operation conditions;Low cost of investment and of operation.

The accuracy of the flow meter also depends on external factors, such as the physical and chemical properties of the medium, geometric distortions of the pipeline, sediment inside of the pipeline and armature fittings disturbing the flow. For that reason, there are many limitations in the use of instruments of a certain type in given conditions which deviate from the standard conditions accepted and described in normative acts (norms, instructions). This was the reason for the authors of this article to perform a comprehensive study connected to the use of the ultrasonic method of measurement in customized measurement conditions, i.e., behind different elements which disturb the flow [6,7,8,9]. Hydraulic elbows of different sizes and shapes, as well as a throttle, were the obstacles in the available literature. In this article, the element considered as being flow disturbing is a knife gate valve—an element used frequently in flow installations.

Ultrasonic flow meters are very popular in measurement practice. Ultrasonic flow meters with clamp-on heads are very versatile because the process of installing them on the outer surface of the pipeline does not interfere with the functioning of the system and does not force the stopping of the system’s operation. Such system stoppages conducted in order to install measuring devices are problematic for their owners, because production stoppages result in financial losses. In some industries, the stopping and restarting of the system is a complex technological process (i.e., stopping and restarting the operation of a power unit). In these sort of situations, to—among others—control the correctness and validity of indications shown by flow meters installed in the system, the usage of ultrasonic flow meters becomes the simplest as well as the cheapest and frequently the most accurate way to perform a measurement of a flow stream. In the case of industrial measurements, it is crucial that the equipment used ensures the required measurement accuracy within a specified period, the longest possible operating time, which reduces the capital expenditure connected to the replacement of the system’s measuring devices. In this aspect, ultrasonic flow meters are characterized by the consistency of their properties and measurement characteristics, which results in the popularity and generality of their applications. The universality of the ultrasonic method of measurements in the scope of the structure of the medium which is being measured is also a significant advantage. Thanks to the use of the ultrasonic method, it is possible to measure single- as well as two-phase flows [10]. Two-phase flows occur frequently in power engineering (the flow of wet steam in a pipeline leading to a turbine), water management (partial contamination of a liquid with a presence of other substances in a sewage treatment plant), in the chemical/food industry (the flow of beet juice/syrup in a sugar factory or flow of a yogurt with fruit pieces) or in the petrochemical industry (mixture of petroleum fractions, mixture of petroleum fraction with air).

The usage of the ultrasonic method of flow measurements has some limitations as well. Ultrasonic flow meters are sensitive to flow disturbances, which—as was mentioned earlier—can be caused by elements of armature fittings or sediment on the pipeline walls. As a result, there is a necessity for the preservation of straight sections of a pipeline in front of and behind the obstacle which is the source of the flow disturbances. The required straight sections of pipelines, the use of which removes the influence of the disturbance caused by the obstacle on the measurement result, are strictly defined for different types of disturbances in the standards and manuals of the devices. The preservation of the above-mentioned straight sections in front of and behind the source of the disturbance, in the case of large installations and pipelines with big diameters, is a real difficulty. In these cases, to compensate for the measurement error caused by the flow disturbances, it is necessary to introduce a correction factor, K*. Determination of the factor is based on the knowledge or adjustment with a sufficiently high accuracy, the equation of the velocity distribution describing a given flow. The formulas describing the actual flow velocity distribution can be adjusted in accordance with theoretical velocity distribution equations [11]. The equations presented in [11] are commonly and often used by authors of articles on ultrasonic flow meters [12,13,14]. This can also be achieved with the use of mathematical modelling. Modelling of flow aspects is frequently performed through CFD software [15,16,17,18]. The measurement method which enables specifying the point velocity values, determining the character of the flow and identifying the velocity distribution is laser anemometry [19].

## 2. Measurements

The purpose of this article was to determine errors in the velocity measurement behind the knife gate valve for two different flow rates (Re = 35,000 and Re = 70,000). The range of the Reynolds number, Re, under examination enabled the verification of the universality of observations and conclusions drawn on the basis of the measurement results analysis. Average speed values for a particular series amounting to v = 0.93 m/s and v = 1.78 m/s, respectively, corresponding to the Reynolds number values, are the velocity levels found in numerous industrial installations. This indicates the practical aspect of the conducted research as well as the possibility of the implementation of the results and measurement requirements into measurement practice. Efforts to determine the errors of ultrasonic flow meters’ indications in non-standard measurement conditions have been made by many researchers [20,21]. Aspects taken into consideration were different methods of mounting the flow meters’ heads along with the mounting accuracy [22,23,24], the influence of various types of elements disturbing the flow [25,26,27,28] as well as the impact of factors such as the temperature [29,30] or fluid pollution [31].

This study was performed for two different levels of closing of the knife gate valve: 1/3 closure of the knife gate valve’s height and 1/2 closure of the knife gate valve’s height (Figure 1).
(1)$$P_{\frac{1}{3}}=\frac{1}{2}\cdot {\pi r}^{2}+2\cdot \int_{0}^{18.63}\mathrm{dx}\int_{8.33}^{33.33-\sqrt{25-x^{2}}}\mathrm{dy}=\frac{1}{2}\cdot \pi \cdot 25^{2}\cdot \int_{0}^{18.63}\mathrm{dx}\int_{8.33}^{33.33-\sqrt{25-x^{2}}}\mathrm{dy}$$
(2)$$P_{\frac{1}{2}}=\frac{1}{2}\cdot {\pi r}^{2}+2\cdot \int_{0}^{21.65}\mathrm{dx}\int_{0}^{25-\sqrt{25-x^{2}}}\mathrm{dy}=\frac{1}{2}\cdot \pi \cdot 25^{2}+2\cdot \int_{0}^{21.65}\mathrm{dx}\int_{0}^{25-\sqrt{25-x^{2}}}\mathrm{dy}$$

The calculations made (1) and (2) allowed for the obtaining of the following results:With the valve placed in the position of closure of 1/3 of the knife gate valve’s height, one obtains P_1/3_ = 78.09% of the active flow area;With the valve placed in the position of closure of 1/2 of the knife gate valve’s height, one obtains P_1/2_ = 60.89% of the active flow area.

The measuring station parameters and other information concerning the way of performing the measurements are summarized in Table 1. Two types of transit-time flow meters with the same accuracy of indication were used for the measurements to determine the influence of the disturbances on the indications of flow meters with the same metrological properties [32,33]. At the straight section of the pipeline in front of the knife gate valve at a required distance from the disturbance, a Micronics PortaFlow 330 flow meter was mounted in a fixed position. The heads of both of the flow meters described below were installed in a V-shaped system (Figure 2a). The V-shaped head installing system is dedicated to pipelines with small diameters, to enlarge the measurement accuracy thanks to a double pass of the ultrasonic wave. The measurement accuracy is larger than in a Z-shaped system with a singular wave pass, because a double pass of the ultrasonic wave minimizes the influence of possible disturbances in the measured flow space.

The value of the volume flow rate is determined from Equation (3).
(3)$$q_{v}=A\cdot v\cdot K\;$$

The time needed for the waves to propagate inside the fluid from transmitter T1 (point A) downstream to transmitter T2 (point C) is given by Equation (4).
(4)$$t_{1}=t_{A-B}=t_{B-C}=\frac{l}{\left(c+v\cdot \cos\alpha \right)}$$

The transit time of the wave from the transmitter T2 (point C) up to the transmitter T1 (point A) is given by (5).
(5)$$t_{2}=t_{C-B}=t_{B-A}=\frac{l}{\left(c-v\cdot \cos\alpha \right)}$$

The time difference between the upstream and downstream is described by Equation (6).
(6)$$\Delta t=\left(t_{C-B}+t_{B-A}\right)-(t_{B-C}+t_{A-B})=2\cdot \left(t_{2}-t_{1}\right)\;=2\cdot \left(\frac{l}{\left(c-v\cdot \cos\alpha \right)}-\frac{l}{\left(c+v\cdot \cos\alpha \right)}\right)\;\;=\frac{2\cdot l\cdot \left(2v\cdot \cos\alpha \right)}{\left(c-v\cdot \cos\alpha \right)\cdot \left(c+v\cdot \cos\alpha \right)}$$

Transit-time flow meters are based on the measurement of the difference in the transit times of the ultrasonic wave, Δt, which is converted into the value of the velocity, v (7), and volume flow rate, q_v_ (8).
(7)$$v=\frac{c^{2}\cdot 2\cdot \left(t_{2}-t_{1}\right)}{4\cdot l\cdot \cos\alpha }=\frac{c^{2}\cdot \left(t_{2}-t_{1}\right)}{2\cdot l\cdot \cos}$$
(8)$$q_{v}=\frac{c^{2}\cdot \left(t_{2}-t_{1}\right)\cdot K\cdot \pi \cdot D^{2}}{8\cdot l\cdot \cos\alpha }=\frac{c^{2}\cdot \left(t_{2}-t_{1}\right)\cdot \pi \cdot D^{2}}{8\cdot \frac{L}{2\cos\alpha }\cdot \cos\alpha }=\frac{c^{2}\cdot \left(t_{2}-t_{1}\right)\cdot \pi \cdot D^{2}}{4L}$$

The above derivation (3)–(8) was made assuming a uniform distribution of velocities in space, for which the shape factor of the velocity distribution is K = 1 (Figure 2a). The research presented in this article was carried out for a disturbed velocity distribution. The flow disturbance was caused by the operation of the knife gate valve. In such cases, it is necessary to determine the correction factor, K*. The experimentally determined values of the K* factor are presented in Section 3 of this article (Results).

A PortaFlow 330 flow meter was used as a reference flow meter. The indications of an Endress+Hausser Prosonic Flow 93T flow meter mounted behind the knife gate valve were compared to the indications provided by the PortaFlow 330 flow meter. During measurements, the distance between the Prosonic Flow 93T and the valve was increased systematically by the distance of the pipeline diameter, D, in the range of 3D–15D measurement cross sections. (Figure 2.)

Performing measurements in cross sections 0D–2D was impossible because of geometrical limitations which make it impossible to install the measuring heads. Within the measurement, in one measurement cross section, the position of the heads of the Prosonic Flow 93T flow meter was altered within the angle scope, amounting to α 0–360° (Figure 3) by the gradual rotation of the heads by 30° around the horizontal axis of the pipeline.

Apart from the main research with the use of ultrasonic flow meters, auxiliary measurements were conducted by means of the laser anemometry method (LDA). The parameters of the measurement devices are presented in Table 2. Thanks to the LDA experiments, the speed distribution at specified distances from the disturbance and at different cross sections (corresponding to measurement cross sections from the ultrasonic tests) was determined. As a result, the analysis was performed taking into account the changes in the speed distribution at the 3D–15D distance range from the knife gate valve. It also enabled the identification of the space within which the flow is being stabilized.

## 3. Results

The results of the measurements performed according to the procedure described in Section 2 are shown below. The research was conducted with the use of measurement devices with the specification presented in Table 2. A PortaFlow 330 flow meter was used as a reference flow meter to register the values of the speed, v_ref,_ at a straight section of the pipeline in front of the knife gate valve. The Endress+Hausser Prosonic Flow 93T flow meter was used to measure the velocity, v_mes,_ in the area of the disturbed flow behind the knife gate valve. A subsequent series of measurements were run in following distances from the knife gate valve in the range of 3D–15D.

The average values of the velocity measured in front of the valve, v_ref,_ and behind the valve, v_mes,_ were calculated for each measurement series. In order to determine an actual dimensionless factor describing the level of flow distortion, a ratio of the average speeds, v_ref_ and v_mes,_ was used (9). The K* factor is a dimensionless parameter reflecting changes in the velocity distribution on the section between the ultrasonic sensors. The K* factor is used by the authors in many articles on ultrasonic flow measurement [7,9,11,12].
(9)$$K^{*}=\frac{v_{\mathrm{ref}}}{v_{\mathrm{mes}}}$$

### 3.1. Measurement Results—½ Closure of the Knife Gate Valve’s Height, Re = 35,000

The values of the velocity registered during the measurements, marked in the graphs (Figure 4, Figure 5, Figure 6, Figure 7 and Figure 8), were used to determine the value of the K* factor. The values of the K* factor calculated for each measurement series were marked in the graphs (Figure 9).

### 3.2. Measurement Results—½ of Closure of the Knife Gate Valve’s Height, Re = 70,000

The values of the velocity registered during the measurements, marked in the graphs (Figure 10, Figure 11, Figure 12, Figure 13 and Figure 14), were used to determine the value of the K* factor. The values of the K* factor calculated for each measurement series were marked in the graphs (Figure 15).

### 3.3. Measurement Results—1/3 of the Knife Gate Valve’s Height Closed, Re = 35,000

The values of the velocity registered during the measurements, marked in the graphs (Figure 16, Figure 17, Figure 18, Figure 19 and Figure 20), were used to determine the value of the K* factor. The values of the K* factor calculated for each measurement series were marked in the graphs (Figure 21).

### 3.4. Measurement Results—1/3 of the Knife Gate Valve’s Height Closed, Re = 70,000

The values of the velocity registered during the measurements, marked in the graphs (Figure 22, Figure 23, Figure 24, Figure 25 and Figure 26), were used to determine the value of the K* factor. The values of the K* factor calculated for each measurement series were marked in the graphs (Figure 27).

### 3.5. Comparison of Results from All of the Measurement Series

Comprehensive charts were created for certain measurement cross sections: 3D, 6D, 9D, 12D and 15D to formulate conclusions concerning the influence of the extent of the valve’s closure and flow rate/Reynolds number on the results of the flow velocity measurement performed with the use of an ultrasonic flow meter at a specified distance behind the valve. Data from all of the measurement series performed at a specified distance from the valve are presented in the graphs below (Figure 28, Figure 29, Figure 30, Figure 31 and Figure 32).

Graphs showing the dependencies of the K* factor (α) (Figure 28, Figure 29, Figure 30, Figure 31 and Figure 32) enable noticing the tendency concerning the shaping of the K* factor value for subsequent distances, D, from the valve. In the measurement cross sections (3D—Figure 28) located closest to the knife gate valve, the largest flow disturbances occur resulting from the opening of the valve. This is the reason why the results of measurements carried out in this area are burdened with the biggest error. The values of the K* factor are the largest in the area closest to the valve and show fluctuations depending on the angle of mounting of the ultrasonic heads, α. In subsequent measurement cross sections (6D—Figure 29 and 3D—Figure 30), gradual flow stabilization occurs, which results in the decrease in the K* factor value. The K* factor in measurement cross sections 6D and 9D shows much smaller fluctuations in the values depending on the angle of the mounting of the heads, α, than in measurement cross Section 3D. In measurement cross sections 12D (Figure 31) and 15D (Figure 32), the set values of the factor are clearly stabilized and fall within the range (1.00–1.03) for the 12D cross section and (1.00–1.02) for the 15D cross section. The percent amplitude of the K* factor values falls in the scope of the flow meter limit error, Δg = +/−2%. This indicates the possibility of performing the measurement of the velocity with an ultrasonic flow meter at these distances without the necessity of choosing an optimal angle for the mounting of the ultrasonic heads.

### 3.6. Treatment of Results of Laser Anemometry Test

Analysis of the results presented above (Section 3.1, Section 3.2, Section 3.3, Section 3.4 and Section 3.5) of the point velocity measurement enabled the drawing of conclusions regarding the optimal location to perform the measurement within the area of the disturbed flow behind the knife gate valve with two levels of closure: closure at 1/3 of the valve’s height and closure at ½ of the valve’s height. To confirm the conclusions formulated on the basis of graph analysis (Figure 4, Figure 5, Figure 6, Figure 7, Figure 8, Figure 9, Figure 10, Figure 11, Figure 12, Figure 13, Figure 14, Figure 15, Figure 16, Figure 17, Figure 18, Figure 19, Figure 20, Figure 21, Figure 22, Figure 23, Figure 24, Figure 25, Figure 26, Figure 27, Figure 28, Figure 29, Figure 30, Figure 31 and Figure 32), tests with the use of a laser anemometer (LDA) were conducted. Measurements were run in cross sections at the distances 4D, 7D, 10D and 15D from the valve. The above-mentioned distances were chosen in accordance with the geometrical limitations connected to the use of a laser anemometer with a tracking system. The results of the measurements run with the use of the anemometer allowed for the creation of profiles of the velocity distribution at different distances from the knife gate valve (Figure 33, Figure 34, Figure 35 and Figure 36). The velocity profiles which were created make it possible to observe alterations in the velocity distribution caused by the knife gate valve.

While observing alterations in the velocity distribution at the distances 4D, 7D, 10D and 15D in the graphs (Figure 33, Figure 34, Figure 35 and Figure 36), a clearly increasing tendency in the levelling of the velocity profiles with the increase in the distance from the valve can be noticed. The velocity profiles showed the largest disturbances in the speed distribution at the smallest distance from the valve—4D. The maximum velocity of the velocity profiles at the 4D distance from the valve (Figure 33a, Figure 34a, Figure 35a and Figure 36a) is clearly shifted down in relation to the profile center. The difference in the velocity values between the area of decreased speed (upper part of the profile) and the area of increased speed (bottom part of the profile) is very visible. At the distance of 7D (Figure 33b, Figure 34b, Figure 35b and Figure 36b) and at the distance of 10D (Figure 33c, Figure 34c, Figure 35c and Figure 36c), the disproportion between the minimal and maximum values of the velocity decreases. The area of increased velocity moves upward, and the area of decreased velocity moves downward, with a target to obtain symmetrical velocity distribution. In the profiles created for the 15D distance from the valve (Figure 33d, Figure 34d, Figure 35d and Figure 36d), stabilized velocity distributions are visible. Changes in the velocity distribution, which are the effect of the presence of the obstacle in the form of a knife gate valve, are minor at the 15D distance. Point differences in the speed values are also little and do not cause measurement errors going over the scope of a limit error of the flow meter used within the ultrasonic measurement which is presented in the graphs (Figure 4, Figure 5, Figure 6, Figure 7, Figure 8, Figure 9, Figure 10, Figure 11, Figure 12, Figure 13, Figure 14, Figure 15, Figure 16, Figure 17, Figure 18, Figure 19, Figure 20, Figure 21, Figure 22, Figure 23, Figure 24, Figure 25, Figure 26 and Figure 27).

While comparing the velocity profiles prepared for the same values of the Reynolds number, Re, larger disorders in the velocity distributions were observed at a bigger level of the valve’s closure—½ of the valve’s height (Figure 33 and Figure 34). While comparing the velocity profiles prepared for the same levels of the valve’s closures, larger disorders in the velocity distributions were observed for a bigger flow—with a Reynolds number, Re = 70,000 (Figure 34 and Figure 36).

## 4. Conclusions

This article presents the results of research on flow velocity conducted with the use of an ultrasonic flow meter in non-standard conditions, behind a knife gate valve which caused flow disturbances. The analysis of the measurement results allows for the identification of an optimal measurement point at the smaller distance from the knife gate valve than is specified in the standards and recommendations. To obtain this purpose, it is necessary to verify if the value of the K* factor calculated for the measurement data is concluded within the limit error range, Δg ∈ <0.98; 1.02>. Optimal positions specified for individual measurement cross sections for running measurements differed by location (the α angle) depending on the Reynolds number, Re, as well as on the level of the knife gate valve closure.

To summarize the results of the measurements and calculations, it should be noted that it is highly reasoned to perform research with the use of a transit-time ultrasonic flow meter in non-standard conditions behind the disturbance in the form of a knife gate valve. At the minimum 12D distance from the valve, the velocity distribution is stabilized enough to enable measurement in any angular setting of the ultrasonic heads. At the measurement cross section distances smaller than 12D, the influence of the velocity distribution disturbances on the measurement results is significant but it is possible to find such mounting positions (position angles) of the ultrasonic heads for which the measurement error is smaller than the limit error.

What is more, the research results and their analysis presented in this article allow to state that:In the measurement series conducted for the ½ closure of the knife gate valve, much larger flow disturbances occurred than in the measurement series conducted for the 1/3 closure of the valve. These observations were also confirmed by the laser anemometry LDA tests. Graphic analysis of the velocity profiles showed an analogy in the flow disturbance structure for both levels of the closure of the valve.In the course of the correlation of the K* factor (α) for the series with Re = 35,000 and Re = 70,000, analogies can be noticed. It can be assumed that this correlation is universal for Reynolds numbers in the range of the turbulent flow.

Practical recommendations formulated on the basis of the tests carried out are to perform the measurement behind the obstacle in the form of a gate valve at a distance greater than 12D. In the case of geometrical constraints that prevent the measurement at a distance greater than 12D, it is recommended to take measurements closer to the obstacle at multiple angles, and then determine the average velocity value. The research can also be performed using a multi-path ultrasonic flow meter. This will make the measured value less sensitive to flow disturbances. This method has been described in many scientific articles [34,35,36,37].

## Figures and Tables

**Figure 1 sensors-23-04677-f001:**
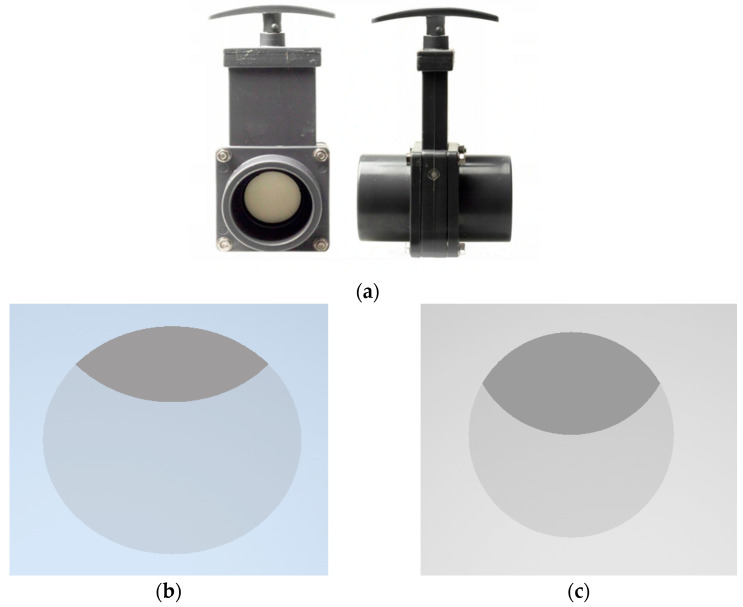
(**a**) Wafer knife gate valve PVC DN50, (**b**) valve closed in 1/3 of the valve’s height, (**c**) valve closed in 1/2 of the valve’s height.

**Figure 2 sensors-23-04677-f002:**
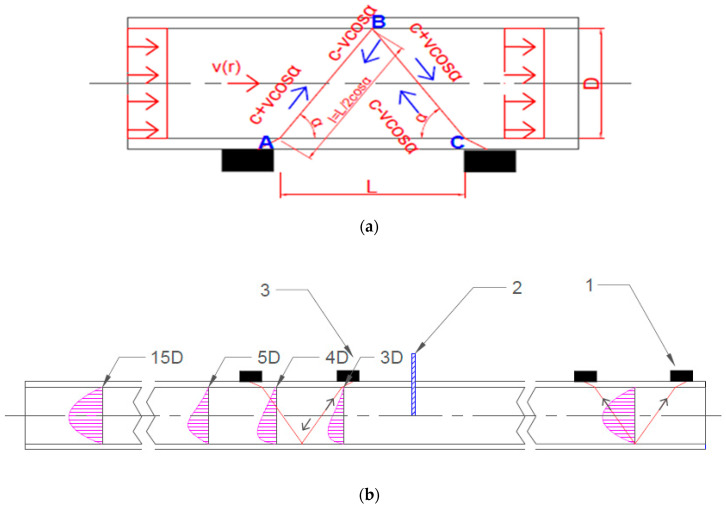
(**a**) Scheme of the ultrasonic wave transition in the system of V-shaped heads. (**b**) Measurement system diagram: 1—ultrasonic flow meter (reference flow meter) Micronics PortaFlow 330, 2—knife gate valve DN50, 3—ultrasonic flow meter Endress+Hausser Prosonic Flow 93T, 3D–15D—measurement cross sections.

**Figure 3 sensors-23-04677-f003:**
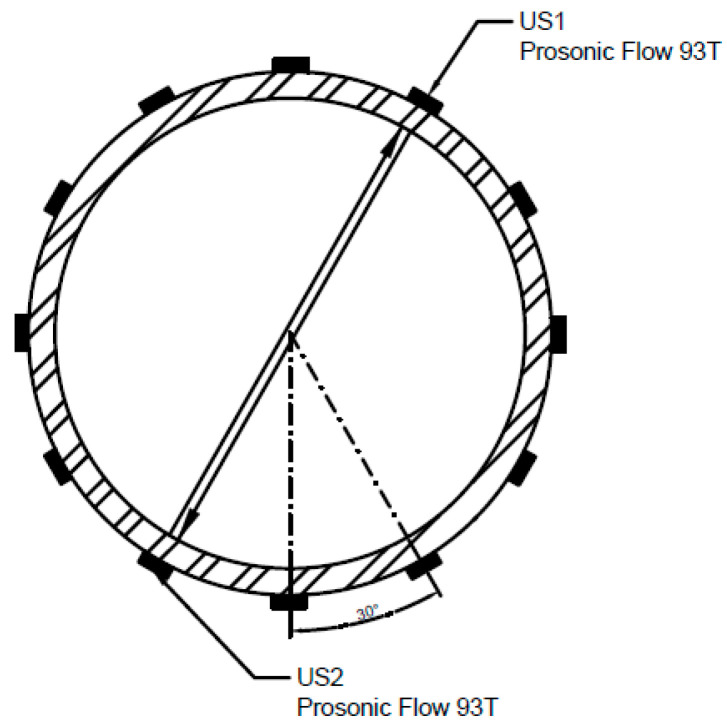
Positions (angles) of the flow meter installation behind the valve at individual 3D–15D measurement cross sections.

**Figure 4 sensors-23-04677-f004:**
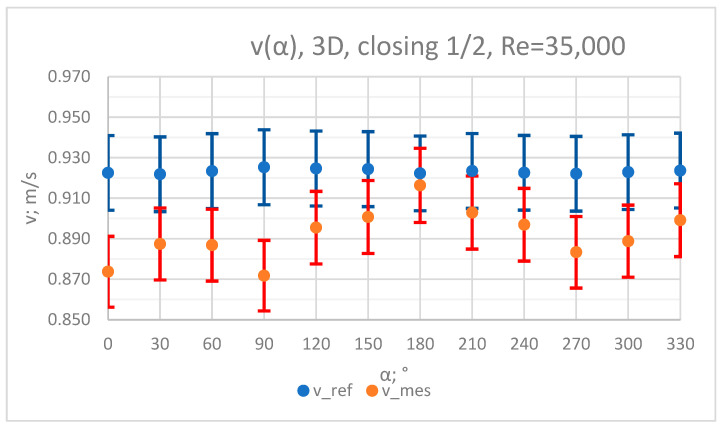
Graph showing values of velocity measured in front of the valve, v_ref,_ and at 3D distance from the valve, v_mes,_ with ½ of the knife gate valve’s height closed and Reynolds number, Re = 35,000.

**Figure 5 sensors-23-04677-f005:**
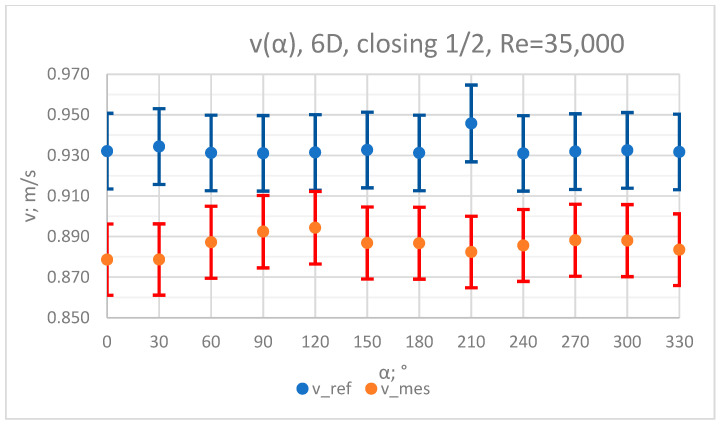
Graph showing values of velocity measured in front of the valve, v_ref,_ and at 6D distance from the valve, v_mes,_ with ½ of the knife gate valve’s height closed and Reynolds number, Re = 35,000.

**Figure 6 sensors-23-04677-f006:**
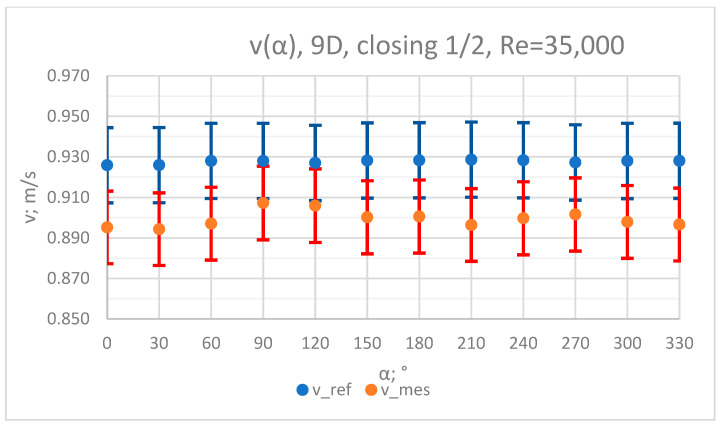
Graph showing values of velocity measured in front of the valve, v_ref,_ and at 9D distance from the valve, v_mes,_ with ½ of the knife gate valve’s height closed and Reynolds number, Re = 35,000.

**Figure 7 sensors-23-04677-f007:**
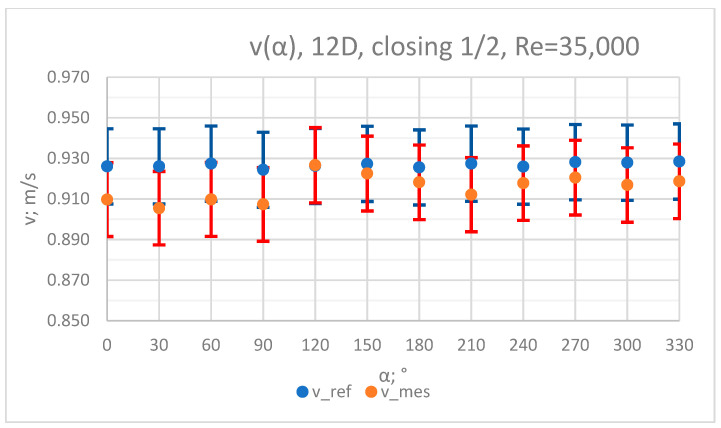
Graph showing values of velocity measured in front of the valve, v_ref,_ and at 12D distance from the valve, v_mes,_ with ½ of the knife gate valve’s height closed and Reynolds number, Re = 35,000.

**Figure 8 sensors-23-04677-f008:**
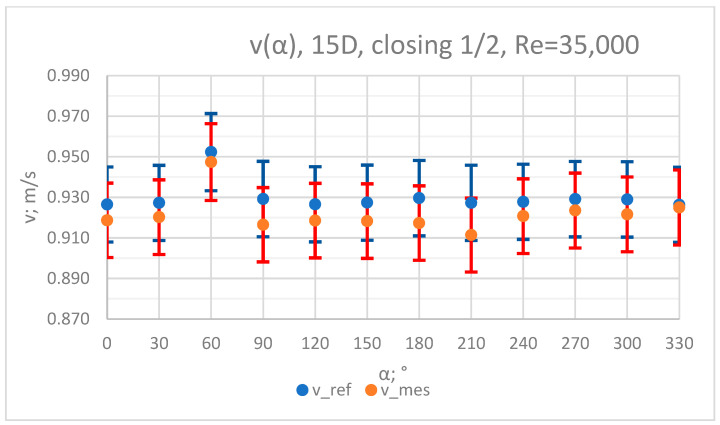
Graph showing values of velocity measured in front of the valve, v_ref,_ and at 15D distance from the valve, v_mes,_ with ½ of the knife gate valve’s height closed and Reynolds number, Re = 35,000.

**Figure 9 sensors-23-04677-f009:**
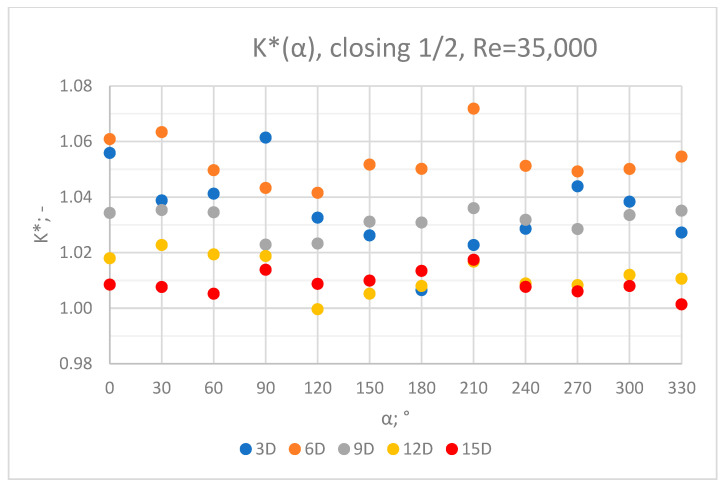
Graph showing values of the K* factor determined for the measured values of velocity, v_ref_ and v_mes,_ with ½ of the knife gate valve’s height closed and Reynolds number, Re = 35,000.

**Figure 10 sensors-23-04677-f010:**
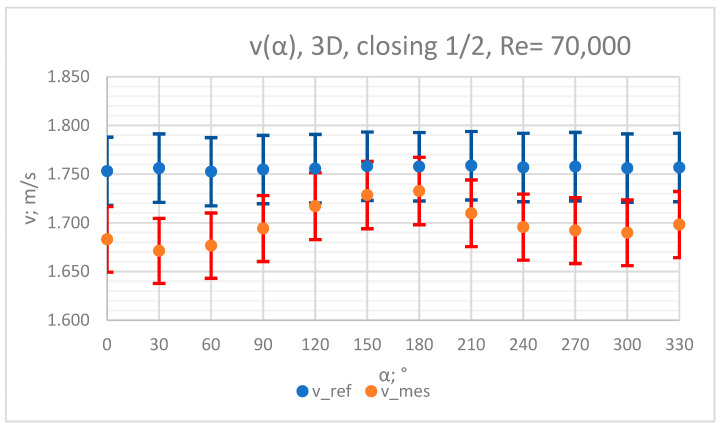
Graph showing values of velocity measured in front of the valve, v_ref,_ and at 3D distance from the valve, v_mes,_ with ½ of the knife gate valve’s height closed and Reynolds number, Re = 70,000.

**Figure 11 sensors-23-04677-f011:**
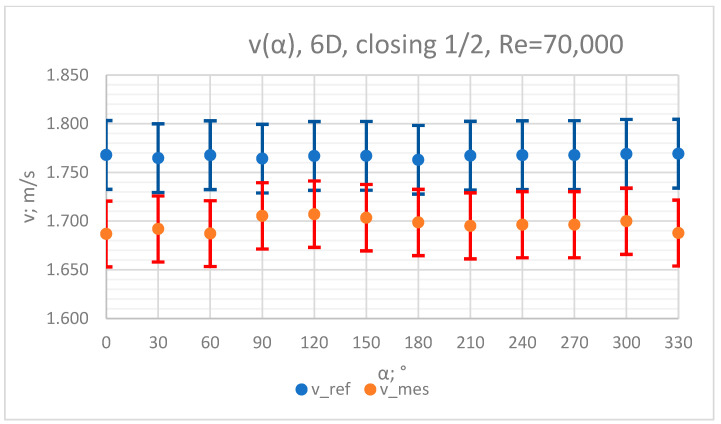
Graph showing values of velocity measured in front of the valve, v_ref,_ and at 6D distance from the valve, v_mes,_ with ½ of the knife gate valve’s height closed and Reynolds number, Re = 70,000.

**Figure 12 sensors-23-04677-f012:**
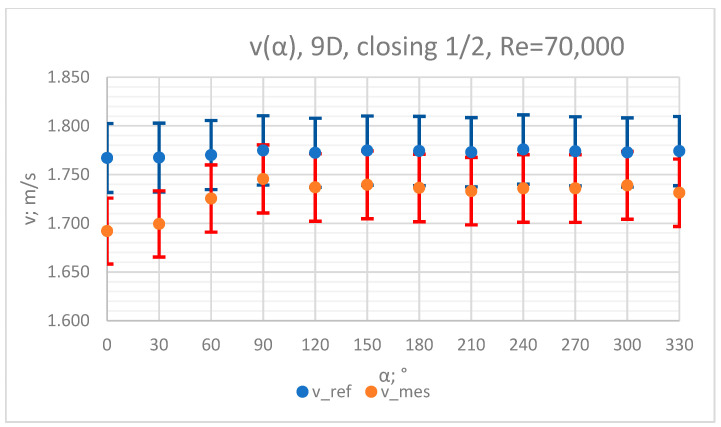
Graph showing values of velocity measured in front of the valve, v_ref,_ and at 9D distance from the valve, v_mes,_ with ½ of the knife gate valve’s height closed and Reynolds number, Re = 70,000.

**Figure 13 sensors-23-04677-f013:**
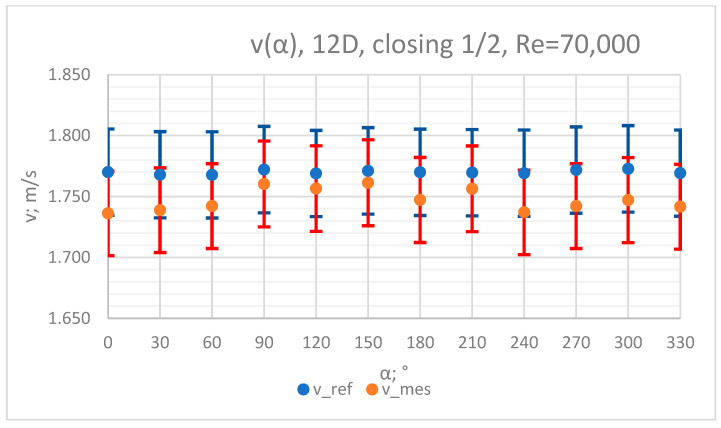
Graph showing values of velocity measured in front of the valve, v_ref,_ and at 12D distance from the valve, v_mes,_ with ½ of the knife gate valve’s height closed and Reynolds number, Re = 70,000.

**Figure 14 sensors-23-04677-f014:**
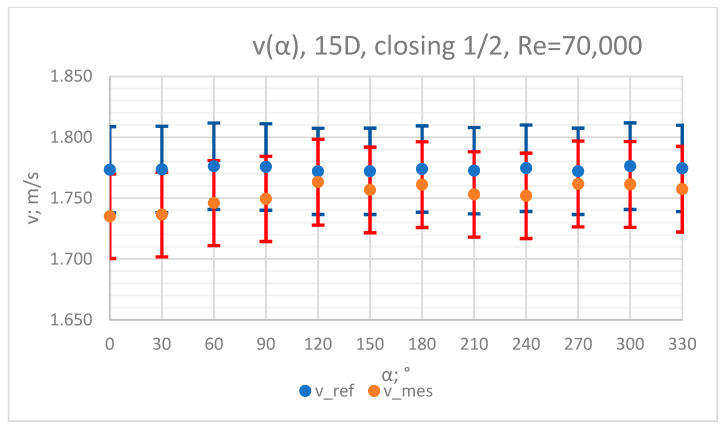
Graph showing values of velocity measured in front of the valve, v_ref,_ and at 15D distance from the valve, v_mes,_ with ½ of the knife gate valve’s height closed and Reynolds number, Re = 70,000.

**Figure 15 sensors-23-04677-f015:**
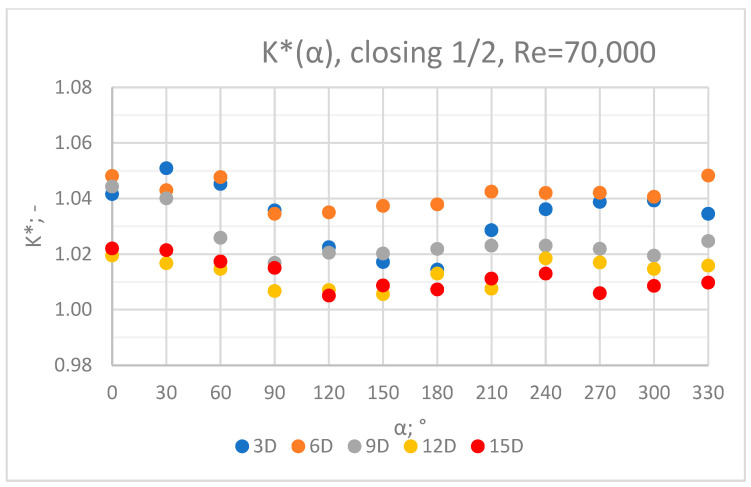
Graph showing values of the K* factor determined for the measured values of velocity, v_ref_ and v_mes,_ with ½ of the knife gate valve’s height closed and Reynolds number, Re = 70,000.

**Figure 16 sensors-23-04677-f016:**
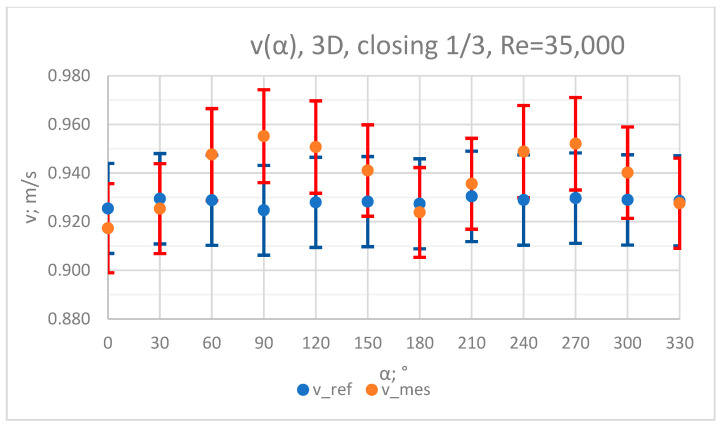
Graph showing values of velocity measured in front of the valve, v_ref,_ and at 3D distance from the valve, v_mes,_ with 1/3 of the knife gate valve’s height closed and Reynolds number, Re = 35,000.

**Figure 17 sensors-23-04677-f017:**
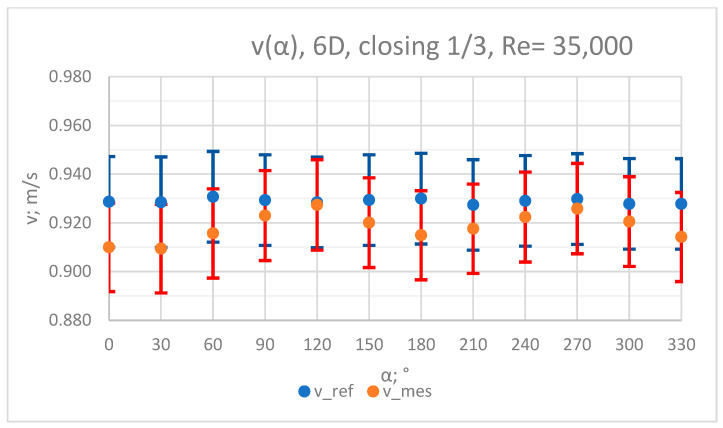
Graph showing values of velocity measured in front of the valve, v_ref,_ and at 6D distance from the valve, v_mes,_ with 1/3 of the knife gate valve’s height closed and Reynolds number, Re = 35,000.

**Figure 18 sensors-23-04677-f018:**
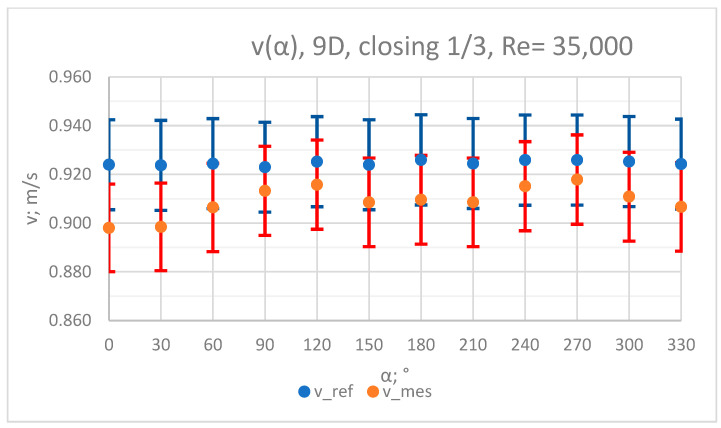
Graph showing values of velocity measured in front of the valve, v_ref,_ and at 9D distance from the valve, v_mes,_ with 1/3 of the knife gate valve’s height closed and Reynolds number, Re = 35,000.

**Figure 19 sensors-23-04677-f019:**
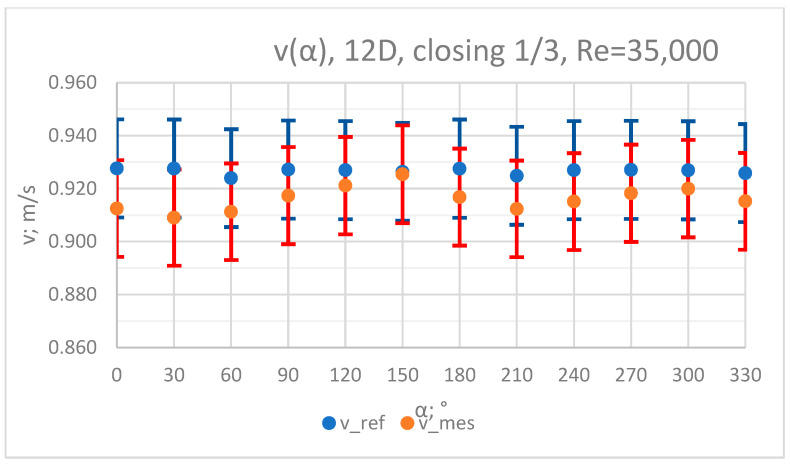
Graph showing values of velocity measured in front of the valve, v_ref,_ and at 12D distance from the valve, v_mes,_ with 1/3 of the knife gate valve’s height closed and Reynolds number, Re = 35,000.

**Figure 20 sensors-23-04677-f020:**
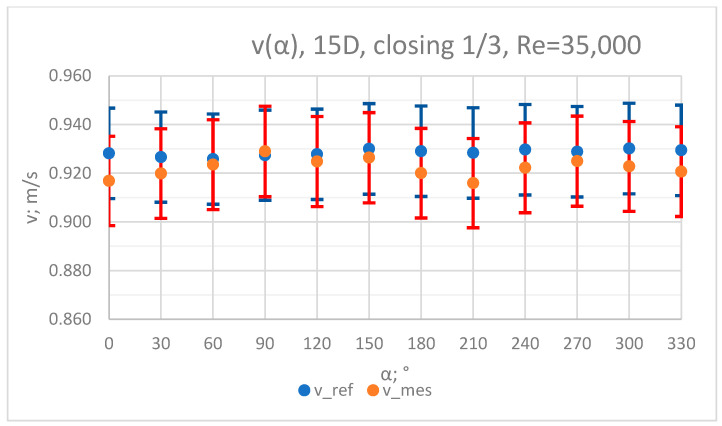
Graph showing values of velocity measured in front of the valve, v_ref,_ and at 15D distance from the valve, v_mes,_ with 1/3 of the knife gate valve’s height closed and Reynolds number, Re = 35,000.

**Figure 21 sensors-23-04677-f021:**
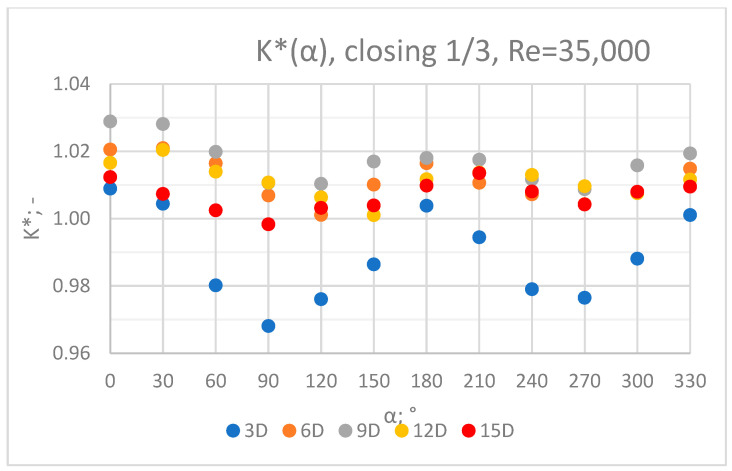
Graph showing values of the K* factor determined for the measured values of velocity, v_ref_ and v_mes,_ with 1/3 of the knife gate valve’s height closed and Reynolds number, Re = 35,000.

**Figure 22 sensors-23-04677-f022:**
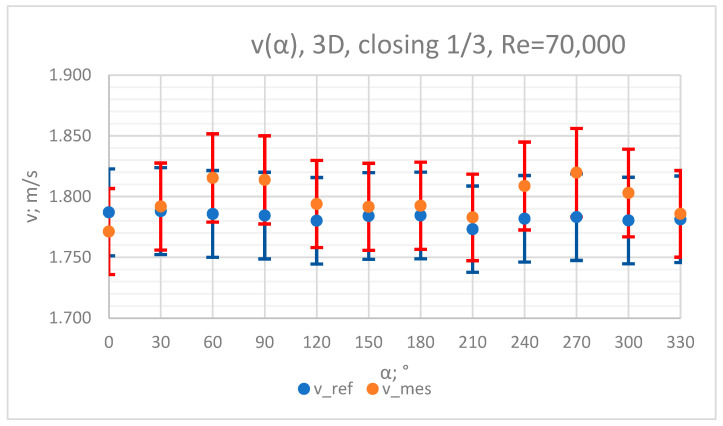
Graph showing values of velocity measured in front of the valve, v_ref,_ and at 3D distance from the valve, v_mes,_ with 1/3 of the knife gate valve’s height closed and Reynolds number, Re = 70,000.

**Figure 23 sensors-23-04677-f023:**
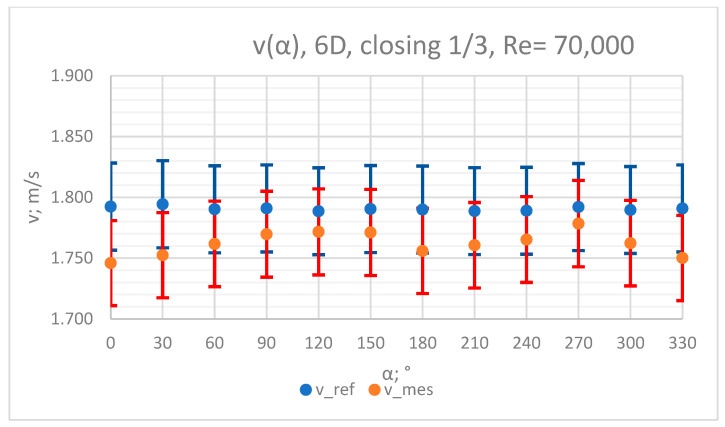
Graph showing values of velocity measured in front of the valve, v_ref,_ and at 6D distance from the valve, v_mes,_ with 1/3 of the knife gate valve’s height closed and Reynolds number, Re = 70,000.

**Figure 24 sensors-23-04677-f024:**
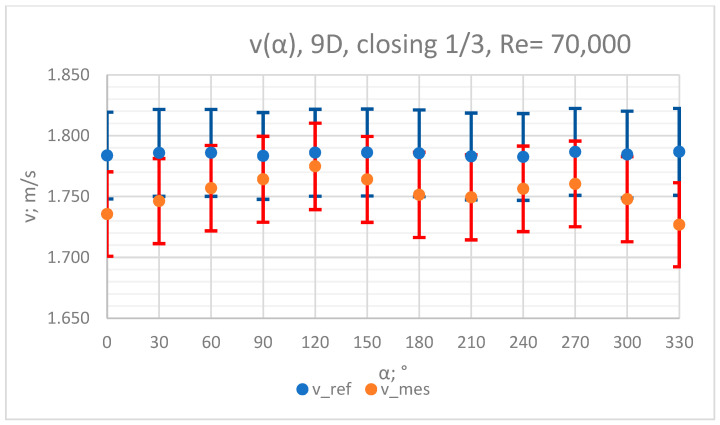
Graph showing values of velocity measured in front of the valve, v_ref,_ and at 9D distance from the valve, v_mes,_ with 1/3 of the knife gate valve’s height closed and Reynolds number, Re = 70,000.

**Figure 25 sensors-23-04677-f025:**
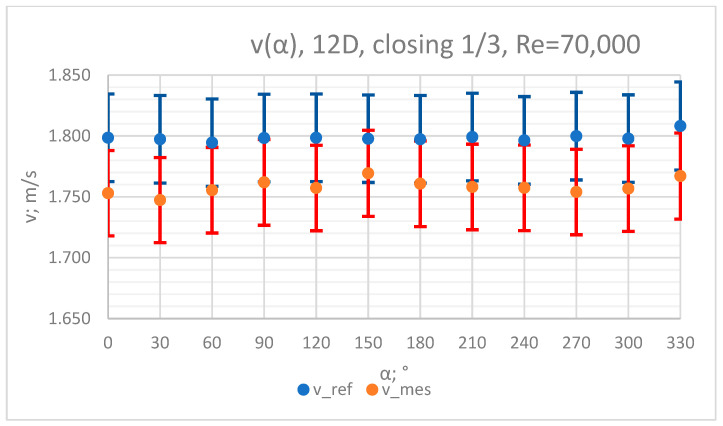
Graph showing values of velocity measured in front of the valve, v_ref,_ and at 12D distance from the valve, v_mes,_ with 1/3 of the knife gate valve’s height closed and Reynolds number, Re = 70,000.

**Figure 26 sensors-23-04677-f026:**
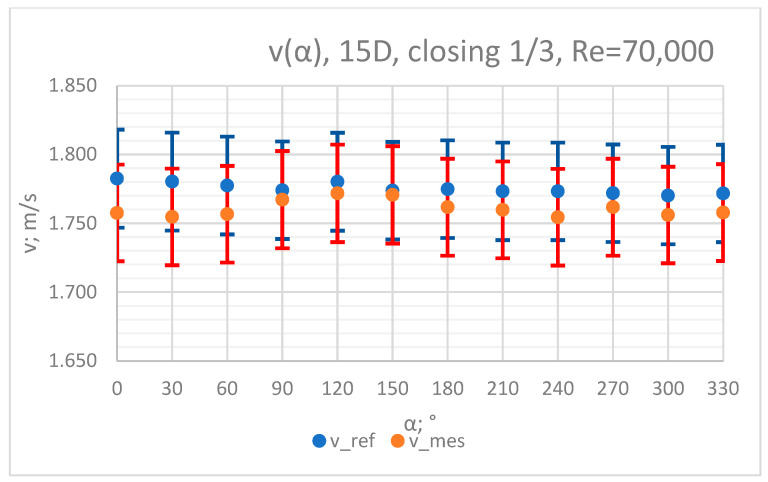
Graph showing values of velocity measured in front of the valve, v_ref,_ and at 15D distance from the valve, v_mes,_ with 1/3 of the knife gate valve’s height closed and Reynolds number, Re = 70,000.

**Figure 27 sensors-23-04677-f027:**
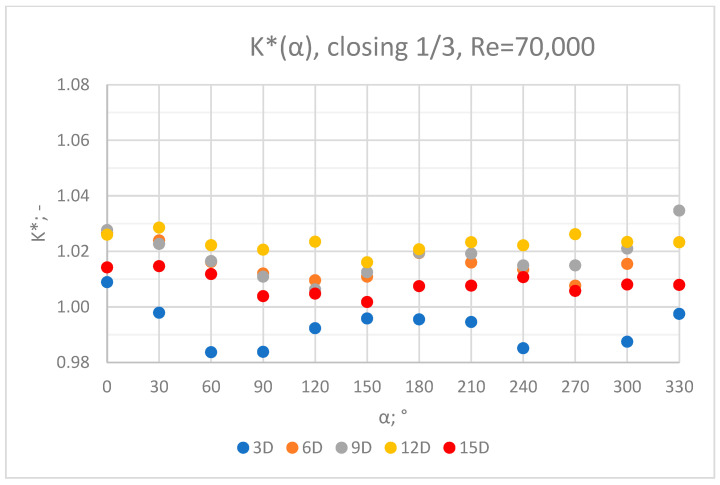
Graph showing values of the K* factor determined for the measured values of velocity, v_ref_ and v_mes,_ with 1/3 of the knife gate valve’s height closed and Reynolds number, Re = 70,000.

**Figure 28 sensors-23-04677-f028:**
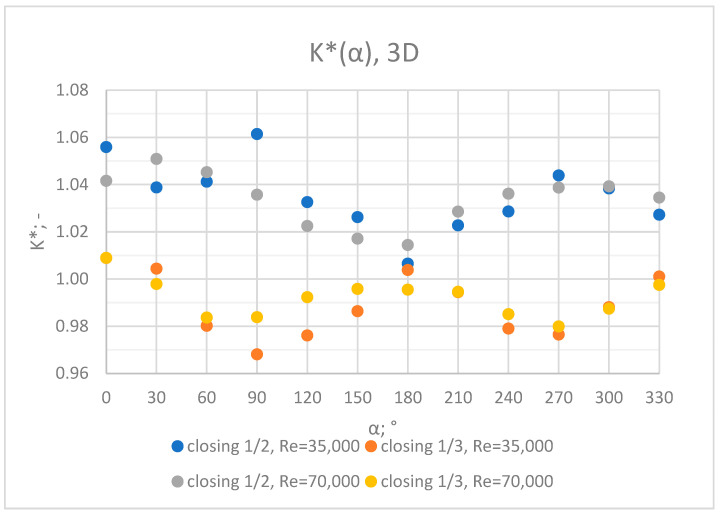
Graph showing values of the K* factor determined for the measured values of velocity, v_ref_ and v_mes,_ at 3D distance with different levels of the knife gate valve’s opening and Reynolds number.

**Figure 29 sensors-23-04677-f029:**
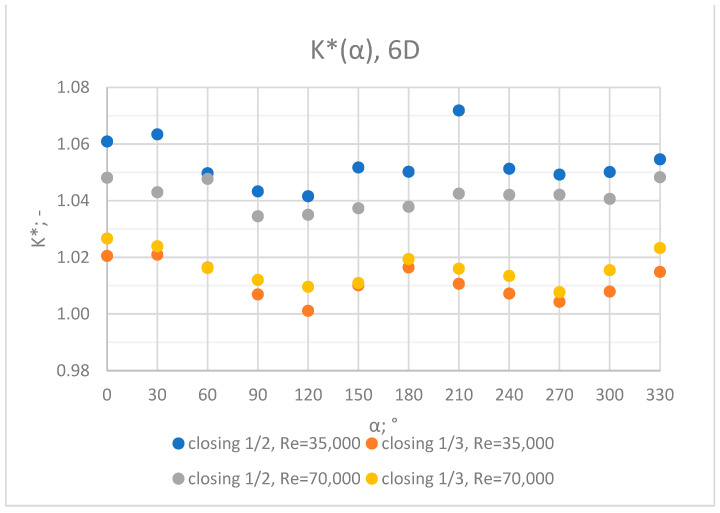
Graph showing values of the K* factor determined for the measured values of velocity, v_ref_ and v_mes,_ at 6D distance with different levels of the knife gate valve’s opening and Reynolds number.

**Figure 30 sensors-23-04677-f030:**
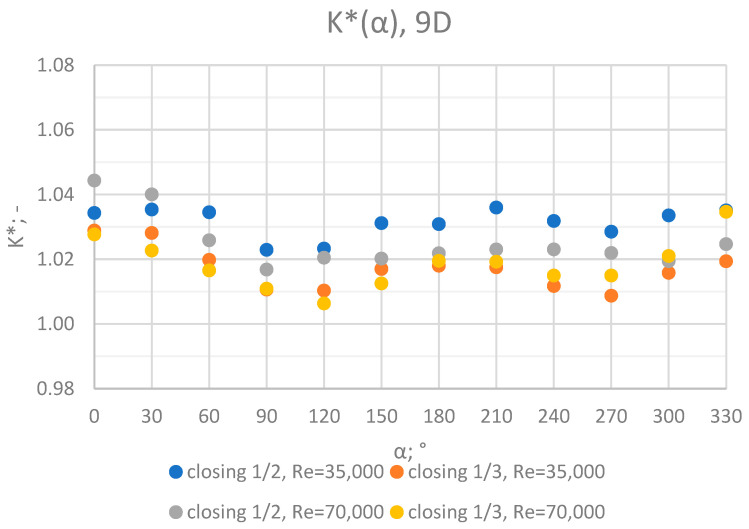
Graph showing values of the K* factor determined for the measured values of velocity, v_ref_ and v_mes,_ at 9D distance with different levels of the knife gate valve’s opening and Reynolds number.

**Figure 31 sensors-23-04677-f031:**
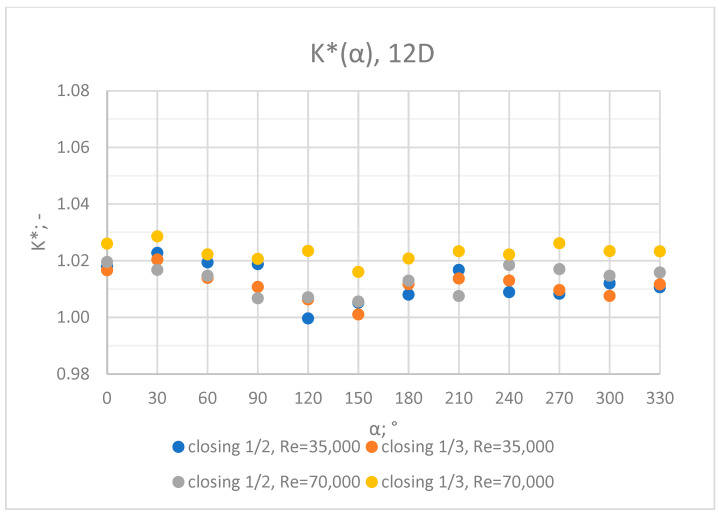
Graph showing values of the K* factor determined for the measured values of velocity, v_ref_ and v_mes,_ at 12D distance with different levels of the knife gate valve’s opening and Reynolds number.

**Figure 32 sensors-23-04677-f032:**
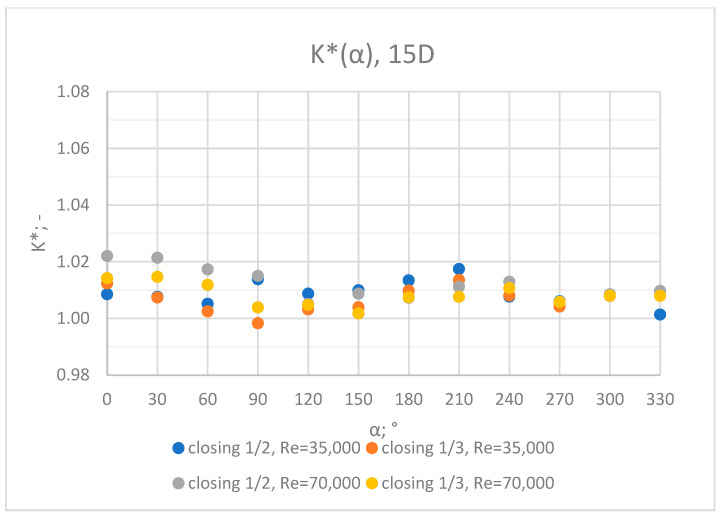
Graph showing values of the K* factor determined for the measured values of velocity, v_ref_ and v_mes,_ at 15D distance with different levels of the knife gate valve’s opening and Reynolds number.

**Figure 33 sensors-23-04677-f033:**
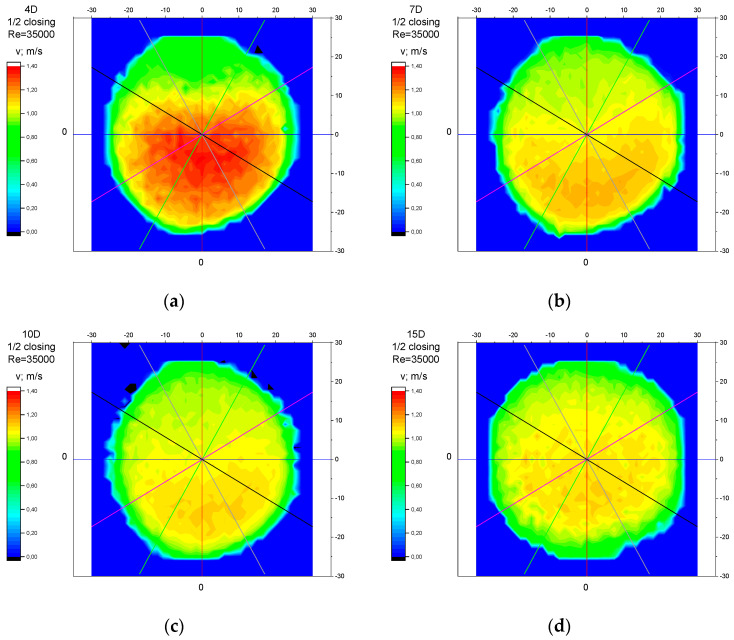
Velocity profiles created on the basis of LDA measurements performed with ½ opening of the knife gate valve and Reynolds number, Re = 35,000, at distances from the valve: (**a**) 4D, (**b**) 7D, (**c**) 10D, (**d**) 15D.

**Figure 34 sensors-23-04677-f034:**
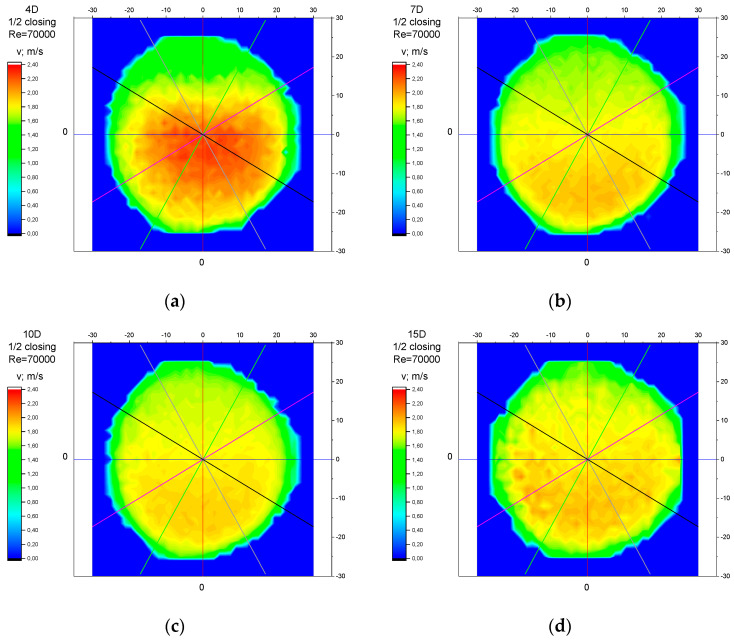
Velocity profiles created on the basis of LDA measurements performed with ½ opening of the knife gate valve and Reynolds number, Re = 70,000, at distances from the valve: (**a**) 4D, (**b**) 7D, (**c**) 10D, (**d**) 15D.

**Figure 35 sensors-23-04677-f035:**
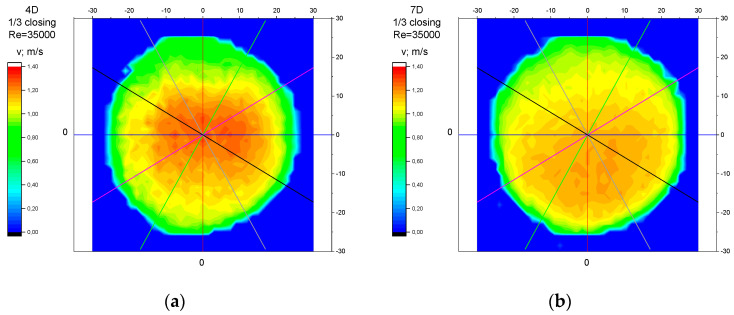
Velocity profiles created on the basis of LDA measurements performed with 1/3 opening of the knife gate valve and Reynolds number, Re = 35,000, at distances from the valve: (**a**) 4D, (**b**) 7D, (**c**) 10D, (**d**) 15D.

**Figure 36 sensors-23-04677-f036:**
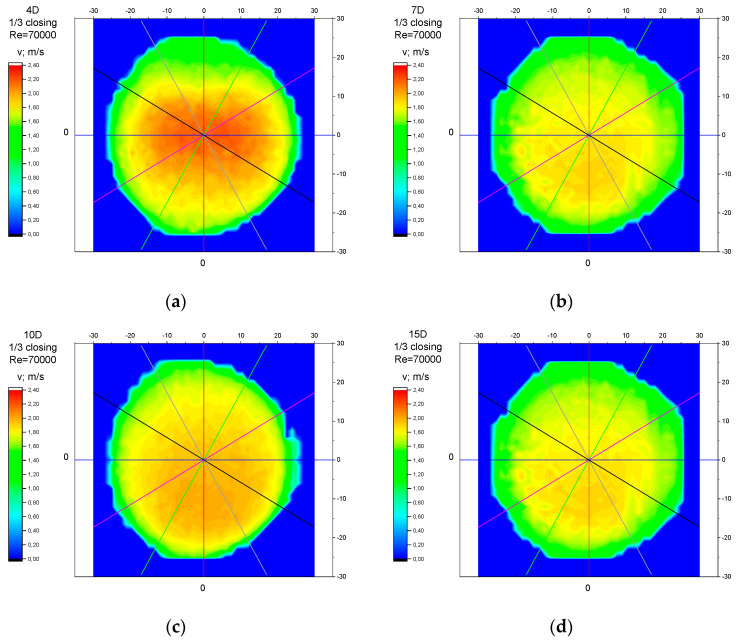
Velocity profiles created on the basis of LDA measurements performed with 1/3 opening of the knife gate valve and Reynolds number, Re = 70,000, at distances from the valve: (**a**) 4D, (**b**) 7D, (**c**) 10D, (**d**) 15D.

**Table 1 sensors-23-04677-t001:** Detailed specification of measurement parameters.

Name of Parameter	Parameter
Nominal diameter of the pipeline	DN 50
Measurement sections	3D–15D
Velocities	ca. 0.3 m/s for Re = 35,000
	ca. 1.78 m/s for Re = 70,000
The ultrasound wave path	V
Sampling interval	5 s
Distance between sensors	ca. 90 mm
Knife gate closing level	1/3 valve height (ca. 22%)
	1/2 valve height (ca. 40%)

**Table 2 sensors-23-04677-t002:** Characteristic values of measuring devices.

Device	Brand	Model	Specification	Maximum Permissible Error (MPE)
Ultrasonic flow meter	Microsonic	Porta Flow 330	Transit-time measurement, clamp-on sensors	0.5% to 2% of velocity reading for v > 0.2 m/s
Ultrasonic flow meter	Endress+Hausser	Prosonic Flow 93T	Transit-time measurement, clamp-on sensors	0.5% to 2% of velocity reading for v > 0.3 m/s for Re > 10,000
Laser Doppler anemometer	Dantec	One-channel laser Doppler anemometer data	Power of laser: 10 mW Light wavelength: 632.8 nm-red Light focal length: 160 mm Measuring volume: 75 μm × 630 μm	-

## Data Availability

All data are integral part of the correspondent author’s Ph.D. thesis.

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
