# Peer review of "Experimental Determination Influence of Flow Disturbances behind the Knife Gate Valve on the Indications of the Ultrasonic Flow Meter with Clamp-On Sensors on Pipelines"

_sensors, 2023, doi:10.3390/s23104677_

Round 1

Reviewer 1 Report (New Reviewer)

Reviewer remarks to the article:

Experimental Determination Influence of Flow Disturbances 3 behind the Knife Gate Valve on the Indications of the Ultra-4 sonic Flow Meter with Clamp-On Sensors on Pipelines

This paper is well constructed. The results of investigations are valuable and very interesting from the point of view of ultrasonic flowmeters in order to improve their accuracy. This is the correct planned and done scientific work. The authors applied a methodical apparatus adequate to the assumed goals. But the following comments should be addressed before considering of publication:

1)     Abstract could be slightly changed, because it is relatively long.

2)     There are some typographical errors in the text.

3)     Please avoid blocks of references, e.g. "...behind different elements which disturb flow [6-9]", as these do not emphasize the particular aspects from the cited papers. Particularly, when citations are made in reference to specific technical aspects, single/double, e.g. [1, 2] references are encouraged. It is strongly suggested that the references need to make in-depth comments on the content of the cited papers; avoid generic comments. Mention/comment the relevance of the cited paper and especially the research gap associated to it.

4)     Make a list of the symbols you use.

5)     Table 2 is twice.

6)     Please unify the drawings; Fig. 2 is hard to read.

7)     The References section is not entirely prepared according to the template.

8)     There is no item 1 in References.

9)     Are items 15 and 16 in References one item?

What can be the indications for industrial practice.

Author Response

Dear Reviewer,

Thanks for all your comments on our article (Manuscript ID: sensors-2331368, “Experimental Determination Influence of Flow Disturbances behind the Knife Gate Valve on the Indications of the Ultrasonic Flow Meter with Clamp-On Sensors on Pipelines”). Below I would like to answer them and clarify your doubts.

Best regards

Piotr Piechota

Reviewer 2 Report (Previous Reviewer 1)

Reference attachment

Author Response

Dear Reviewer,

Thanks for all your comments on our article (Manuscript ID: sensors-2331368, “Experimental Determination Influence of Flow Disturbances behind the Knife Gate Valve on the Indications of the Ultrasonic Flow Meter with Clamp-On Sensors on Pipelines”). I am sending the article with the corrections in the attachment (corrected, shortened abstract, corrected errors in the numbering of chapter 5(4) and in the numbering of figures and references, additional references added, changes made in the conclusions, added list of symbols).

According to the review, the article has been corrected. A separate list of symbols used in the article has been added, as suggested by the reviewer.In accordance with the earlier comments of the reviewer (Reviewers comment 10), in the current version of the article, literature items from recent years have been added [33] -2023 [34] - 2022 [36] -2017. Errors in the numbering of chapters (comment 7) and the correct numbering of citations (comment 10) have been corrected. The derived equations were taken from WaluÅ› S., "Ultrasonic Flowmeters Methodology of Application", Wydawnictwo Politechniki ÅšlÄ…skiej, Gliwice 1997, ISBN 83-85718-43-5 (as a book in Polish, I did not include the source in the article). In the current version of the article, the abstract has been changed for the sake of clarity. The introduction of the article presents a description of applications and characteristics (advantages and disadvantages) of ultrasonic flowmeters. Then attention was paid to the problem of taking measurements behind obstacles, in non-standard measurement conditions. Part 2 presents the derivation of the formula for the measured value of the velocity v (according to the measurement principle) and the work/setting parameters of the flowmeter.

Best regards

Piotr Piechota

Round 2

Reviewer 2 Report (Previous Reviewer 1)

Theoretical formulas, please check carefully. There are some errors, and the article can be published after the change

Author Response

Dear Professor,

Thanks for all your comments on our article (Manuscript ID: sensors-2331368, “Experimental Determination Influence of Flow Disturbances behind the Knife Gate Valve on the Indications of the Ultrasonic Flow Meter with Clamp-On Sensors on Pipelines”). Below I would like to answer them.

Comments for Authors:

Theoretical formulas, please check carefully. There are some errors, and the article can be published after the change

Response:

According to the reviewer's suggestion, the correctness of introducing formulas 4,5,6 was checked. Editing errors made when adding formulas were noted. Changes were made in edit mode.

The missing symbols and notations (c,l,L) have also been supplemented.

1 reference has been added [37].

Sincerely, Piotr Piechota

This manuscript is a resubmission of an earlier submission. The following is a list of the peer review reports and author responses from that submission.

Round 1

Reviewer 1 Report

Please download the review document.

Reviewer 2 Report

The authors performed an experimental study about the readings of an ultrasonic clamp-on flow meter behind a flow obstacle. It is known that obstacles, bends and other deviations from the straight pipe cause misreading in flow meters as certain symmetry assumptions are violated. Therefore, producers typically give minimum L/D ratios for flow meter operation.

I do not see that this article adds anything to the world’s knowledge on flow meters. The work comes like a good master thesis project. It reports on some measurements and the course of a correction factor that may be applied. At its best, it gives a recommendation on potential errors for a specific flwo channel, a specific obstacle, a specific flowmeter, a specific fluid and two Re. This is nothing to be published. I advise the authors to seek explanations and solutions, which are more general, e.g. by considering dimensionless numbers or deriving empirical formulae for the correction factor. However, I know that this is difficult if not impossible at all.

Furthermore, I would like to give the following general comments:

It is not required to have so many plots in a scientific paper. Authors should condense the information in an appropriate way. Their EXCEL-like appearance is also not optimal for scientific publications.

I miss an explanation, why exactly the small Re range (35k...70k) has been chosen and experiments have only been made for two Re. After the obstacle there will be a stagnation zone whose geometry is very much Re dependent.

There is no formulae for K* though in the conclusion a “correlation” is mentioned (second bullet).